# Potential of Viruses as Environmental Etiological Factors for Non-Syndromic Orofacial Clefts

**DOI:** 10.3390/v16040511

**Published:** 2024-03-27

**Authors:** Thiago S. Messias, Kaique C. P. Silva, Thiago C. Silva, Simone Soares

**Affiliations:** 1Hospital for Rehabilitation of Craniofacial Anomalies, University of São Paulo, Bauru 17012-901, SP, Brazil; tsilvamessias@gmail.com (T.S.M.); kaiquecesar@alumni.usp.br (K.C.P.S.); 2Faculty of Medicine, Nove de Julho University, Bauru 17011-102, SP, Brazil; 3Faculty of Architecture, Arts, Communication and Design, São Paulo State University, Bauru 17033-360, SP, Brazil; thiagotkrvalho@gmail.com; 4Department of Prosthodontics and Periodontology, Bauru School of Dentistry, University of São Paulo, 9-75, Bauru 17012-901, SP, Brazil

**Keywords:** viruses, cleft lip, cleft palate, etiology, mosquito-borne viruses, pathogenesis

## Abstract

In this study, we analyzed the potential of viral infections in the species *Homo sapiens* as environmental causes of orofacial clefts (OFCs). A scoring system was adapted for qualitatively assessing the potential of viruses to cause cleft lip and/or palate (CL/P). This assessment considered factors such as information from the literature, nucleotide and amino acid similarities, and the presence of Endogenous Viral Elements (EVEs). The analysis involved various algorithm packages within Basic Local Alignment Search Tool 2.13.0 software and databases from the National Center for Biotechnology Information and the International Committee on Taxonomy of Viruses. Twenty significant viral species using different biosynthesis strategies were identified: *Human coronavirus NL63*, *Rio Negro virus*, *Alphatorquevirus homin9*, *Brisavirus*, *Cosavirus B*, *Torque teno mini virus 4*, *Bocaparvovirus primate2*, *Human coronavirus HKU1*, *Monkeypox virus*, *Mammarenavirus machupoense*, *Volepox virus*, *Souris mammarenavirus*, *Gammapapillomavirus 7*, *Betainfluenzavirus influenzae*, *Lymphocytic choriomeningitis mammarenavirus*, *Ledantevirus kern*, *Gammainfluenzavirus influenzae*, *Betapolyomavirus hominis*, *Vesiculovirus perinet*, and *Cytomegalovirus humanbeta5*. The evident viral etiological potential in relation to CL/P varies depending on the Baltimore class to which the viral species belongs. Given the multifactorial nature of CL/P, this relationship appears to be dynamic.

## 1. Introduction

Cleft lip and/or palate (CL/P) are the most frequent craniofacial malformations [1] causing limitations to patients, especially if not treated in a specialized and multidisciplinary way [2]. CL/P not associated with syndromes have a complex etiology, and among the associated factors are viral congenital infections [3,4].

Viruses are microorganisms with parasitic behavior at the intracellular level commonly associated with diseases in humans; they have also been linked with congenital infections and have been identified as teratogens [5], such as the species *Rubivirus rubellae* (RUBV), which is associated with various anomalies [6]. Unlike other organisms, viruses have seven ways of organizing their genetic information, classified on the Baltimore scale as follows: I—double-stranded deoxyribonucleic acid (dsDNA); II—single-stranded deoxyribonucleic acid (ssDNA); III—double-stranded ribonucleic acid (dsRNA); IV—single-stranded ribonucleic acid with positive polarity (ssRNA+); V—single-stranded ribonucleic acid with negative polarity (ssRNA−); VI—single-stranded ribonucleic acid with positive polarity mediated by reverse transcription; VII—partial double-stranded deoxyribonucleic acid mediated by reverse transcription. The existence of all these forms of genomic conservation means that viral biosynthesis is characterized by unique molecular strategies in each class, but with the common objective of more effectively synthesizing messenger ribonucleic acid (mRNA), which is the basis for the synthesis of structural and non-structural viral proteins to form new virions [7,8]. The host cell is affected in several ways, also in accordance with the Baltimore class to which the virus belongs, with viral evasion, control of cell metabolism, suppression and stimulation of expression of host genes, formation of episomes, and cell transformation being some of the many strategies that facilitate viral replication [5,9].

In the surviving hosts of a viral infection, the fixation of Endogenous Viral Elements (EVEs) may occur; EVEs are sequences integrated into the host genome due to the process of coevolution with the host, acting as a “genomic record of the occurrence of the infection” [10]. EVEs are integrated through non-homologous recombination, retroviral integration, and interaction with cellular retroelements. Infection by viruses of the seven Baltimore classes can result in EVE integration into the host genome, with an estimated 5–8% of the genomes of eukaryotic organisms being composed of EVEs [10,11].

Considering viral survival strategies and possible damage to and coevolutionary interactions with the host cell, it is plausible to hypothesize the possibility of viral participation in the pathogenesis of CL/P (i.e., non-tissue fusion, which may be related to changes in one or more cell signaling pathways responsible for migration, mitosis, apoptosis, and cell differentiation). Therefore, in the present study, we aimed to analyze the performance of viruses as potential etiological agents of CL/P and, if there is such a relationship, to determine with which Baltimore class they can be associated; finally, we aimed to investigate the possibility of a coevolutionary history between viruses and genome sequences of the species *Homo sapiens* associated with CL/P (presence of EVEs). 

## 2. Materials and Methods

Based on the method by Silva, Messias, and Soares (2022) [12], a score-based system was employed for the qualitative assessment of the viral potential to cause orofacial clefts as follows (Table 1): I—Is there evidence in the literature of passage through the placenta? Yes, 2; not investigated/not found, 1; no, 0. II—Are there studies associated with orofacial clefts in the literature? Yes, with evidence based on polymerase chain reaction and/or viral isolation, 3; yes, with serological evidence, 2; yes, with semiological evidence, 1; no/not found, 0. III—Total nucleotide similarity (number of significant matches). IV—Similarity of amino acids (number of significant matches). V—Presence of Endogenous Viral Elements (number of EVEs evidenced). The Total Potential Value (TPV) was calculated as the sum of all numerical values obtained in the topics.

The scores for topics I and II were obtained by performing an integrative analysis of the literature with the SciELO, PubMed/MEDLINE, LILACS, and Google Scholar databases in order to obtain a greater number of results. The following keywords were used as descriptors: “viruses”, “virus”, “cleft lip”, “cleft palate”, “orofacial cleft”, “transplacental passage”, and “congenital infection”. No time filter or limitations regarding the type of study or language were applied, with priority in topic II being given to the method of viral detection from the least specific to the gold standard in terms of viral diagnosis (semiology, serology, viral isolation, and/or detection of viral genetic material). After the initial search phase was completed, initially, the title and abstract of each study were read (by T.S.M. and K.C.P.S.). The articles were read sequentially (by T.S.M. and K.C.P.S), and any conflicts between the examiners were resolved (by T.C.S. and S.S.); the studies relevant to the scoring of topics I and II were then used.

The analysis based on nucleotide similarity (topics III, IV, and V) was planned premised on the intimate coevolution of viruses [10] with their hosts and their consequent information-sharing strategies from both sides. The question was: “There is a significant similarity between viral sequences and human sequences related to orofacial clefts, and would not it be indicative of a current (topic III and IV) and/or ancient (topic V) relationship?” To filter the results of each case, the E-value limits were defined in formatting considering the path gene and/or transcript » protein, and EVEs.

For topic III, the software application Basic Local Alignment Search Tool (BLAST; version 2.13.0) was used [13]. To search for similarity with human sequences associated with orofacial clefts, the genomic + transcript databases (human genomic plus transcript (human G + T)) were selected. Three Entrez queries were used as a filter for human sequences: “Orofacial cleft”, “Cleft lip”, and “Cleft palate”. The mega blast, discontiguous megablast, and blastn programs were run with the statistical threshold E-value adjusted to consider only results ≤1. Sequences of positive matches with E-value ≤ 0.05 were analyzed for amino acid similarity (topic IV), and when there were matches with results ≤0.0001, these sequences were analyzed for the presence of EVEs (topic V).

For topic IV, all positive sequences of topic III with the statistical threshold E-value equal to or less than 0.05 were considered, and when these were found to be coding sequences, the Blastp algorithm of Basic Local Alignment Search Tool software (BLAST; version 2.13.0) was used [13] for protein similarity analysis.

For topic V, only sequences with matches ≤0.0001 (obtained in topic III) were analyzed. For the investigation of EVEs, we adapted the methodology by Katzourakis and Gifford (2010) [11], according to which when a nucleotide sequence is identified as a potential EVE, confirmation must be performed by aligning the sequence with a non-specific genomic database; if, in the alignment, a target virus-matching result is obtained, the sequence is determined to be a possible Endogenous Viral Element. 

The nucleotide and amino acid sequences used during the experiment were retrieved from the public databases of the National Center for Biotechnology Information (NCBI) [14] and International Committee on Taxonomy of Viruses (ICTV) [15]. Genome sequences, transcriptomes, and proteomes related to reference viral species that infect the species *Homo sapiens*, and host sequences filtered with the descriptors orofacial cleft, cleft lip, and cleft palate were used. The identifiers of each viral sequence used are described in Appendix A.

To define which viral species to include in the sample groups (viruses that infect the species *H. sapiens*), we used the reference textbook in the area of human virology, *Fields Virology*, in its sixth [16] and seventh editions [17,18], and the ViralZone platform [8] as a basis for selection. 

To infer the time of fixation of the EVEs, BLASTs [13] of the EVEs sequences were performed using the nucleotide collection database (nr/nt) with the filters of the following taxa: chimpanzees (tax id: 9596), gorillas (tax id: 9592), and orangutans (tax id: 9601). The E-value considered to be a significant result was ≤0.0001, and as a basis for the definition of approximate time, we used the references provided in the study on human speciation time by Pääbo (2003), which presents the speciation of orangutans and the human ancestor 12–16 million years ago, gorillas and the human ancestor 6–8 million years ago, and *H. sapiens* and chimpanzee speciation 4–6 million years ago [19].

For the statistical analysis of the TPV values of each virus analyzed, SPSS statistics 28.0 software was used [20]. The Kruskal–Wallis non-parametric test was performed, followed by Dunn’s test for the multiple comparison of independent groups, aiming to investigate the relationships among Baltimore classes.

## 3. Results

A total of 442 viruses, as described in Appendix A, were analyzed and divided into seven groups (G1–G7) according to the Baltimore classification. Using the Total Potential Value (TPV) of each virus, a descriptive statistical analysis, as presented in Table 2, was performed.

The Kruskal–Wallis non-parametric test showed a statistically significant difference (H = 118.711 with 5 degrees of freedom, *p* ≤ 0.001). Differences between median values were also significant (*p* ≤ 0.001), indicating that group differences were much greater than expected by chance. Group 7 (G7) was excluded from the analysis because it did not have a sufficient sample size, with only one virus being assigned to this group (*Hepatitis B virus* (HBV)) with TPV = 7.

To compare the remaining groups, Dunn’s statistical method was used, as shown below (Table 3).

According to the results analyzed by using Dunn’s test, we found significant differences among the groups, thus presenting a significant relation between Baltimore classes I, III, and V and classes II and IV.

Such a comparison of the seven groups can be observed in the boxplot in Figure 1.

The outliers (red dots in Figure 1) regarding their species, abbreviation, Baltimore class, and TPV are reported in Table 4.

Of the 442 viruses analyzed, 20 had atypical TPVs within their sample groups, 5 from G1, 4 from G2, 4 from G4, and 7 from G5. There were no outliers in sample groups 3, 6, and 7.

In Table 5 and Figure 2, we show the EVEs associated with human sequences related to CL/P.

A possible coevolutionary history was evidenced by fixing EVEs in human sequences filtered to be related to CL/P. All EVEs were from viruses belonging to G1 with the following inferred integration times (based on Pääbo (2003)) [19]:Less than 6 million years ago (EVEs present only in *H. sapiens*): *Akhmeta virus* (AKMV), *Molluscum contagiosum virus* (MCV), and *Roseolovirus humanbeta7* (HHV-7);Between 6 and 8 million years ago (EVEs present in humans and chimpanzees): *Alphapapillomavirus 11* (HPV34) and HHV-7;Over 16 million years ago (EVEs present in humans, chimpanzees, gorillas, and orangutans): *Taterapox virus* (TATV), *Camelpox virus* (CMLV), *Varicellovirus humanalpha3* (HHV-3), *Cytomegalovirus humanbeta5* (HHV-5), *Abatino macacapox virus* (AMV), MCV, and HPV34.

## 4. Discussion

A total of 442 viral species were in silico analyzed at the OMA level (genome, transcriptome, and proteome) to investigate their etiological potential for CL/P. In the presence of a relationship between the virus and the disease, the Baltimore class was determined, and the coevolutionary history was evidenced. Twenty species showed atypical TPVs. Further, G1, G3, and G5 showed a significant difference in relation to G2 and G4. An indirect coevolutionary history associates viruses and CL/P (presence of EVEs), and the connection with herpesviruses, papillomaviruses, and poxviruses may date over sixteen million years ago. 

G4 and G5 had the highest sample numbers (129 and 137, respectively), which corroborates the great diversity of ribonucleic acid (RNA) viruses compared with deoxyribonucleic acid (DNA) viruses, which is due to their high mutation rates (~10^−4^ nucleotide substitutions/replications) [21]. 

G7 (Appendix A) presented only one virus (HBV) and, for this reason, was excluded from the statistical analysis. Considering its peculiar replication strategy, HBV has a partial and circularized dsDNA. The difference between Baltimore classes I and VII is the presence of a pre-genomic ribonucleic acid (pgRNA) portion that undergoes reverse transcription into DNA (class VII) during the virion assembly process [8,22]. So far, HBV is the only virus of this class that is related to infection and has been identified as an etiological agent of diseases in our species [22,23]; however, considering the dynamic viral evolution [10] and that there are still several gaps regarding HBV to be investigated, it is not prudent to neglect its etiological capacity because of the limitation evidenced in this research study, since it shares a genus (*Orthohepadnavirus*) with taxa that infect a wide variety of host species, making the adaptive ability of this Baltimore class in the ecosystem at least dynamic [15,23].

G6 (Appendix A) included *n* = 5, without registering atypical samples or significant characters in Dunn’s tests. We also did not find evidence of the integration of any EVE associated with CL/P, which disagrees with the general data presented by Katzourakis and Gifford (2010) and Geoghegan and Holmes (2020), who indicated retroviruses (members of Baltimore class VI) as the main fixators of EVEs in nature [10,11] due to provirus-generating biosynthesis and several other host DNA manipulation mechanisms [24].

In contrast, G5 (Appendix A) had the largest sample size, *n* = 137, with an average TPV = 27 (Table 2). Dunn’s test revealed a significant difference in relation to G4 and G2 (Table 3); therefore, we can infer that some factor in the biosynthesis strategy of ssRNA− viruses is significant to the etiological potential if associated with ssRNA+ and ssDNA. Without going into the peculiar details of certain viral species and genera, Baltimore class V viruses use their template genome (ssRNA−) as the basis for replication, initiated by the viral enzyme RdRp (RNA-dependent RNA polymerase), which transcribes positive-sense strands, which, in turn, are translated into non-structural proteins (replication and viral evasion enzymes) and structural proteins (capsid, envelope proteins, etc.). Used as a viral factory, cell transport pathways and genome replication sites vary according to viral species and infected cell types [25]. This group was not found to fix EVEs, which can be explained considering that RNA viruses without the Reverse Transcriptase enzyme fix EVEs using cellular retroelements, and these occurrences are not as frequent as in retroviruses [10]. However, 7 viruses out of 137 showed atypical TPVs, making G5 the predominant sample group in Table 4. Remembering that outliers are defined as results that differ significantly from the group to which they belong [26], we can demonstrate that they have a greater etiological potential for CL/P than the other viruses analyzed (Figure 1).

Three of the seven G5 outliers belong to the genus *Manmarenavirus*: *Mammarenavirus machupoense* (MACV), *Souris mammarenavirus* (SOUV), and *Lymphocytic choriomeningitis mammarenavirus* (LCMV) [15]. MACV had a TPV of 86, where 61 points of this total came from the matches at the protein level, including the nucleocapsid protein, which has the structural function of accommodating and protecting viral genomic and antigenomic RNA, in addition to the non-structural function of exoribonuclease (RNA degradation) and interferon antagonization (inactivation of the antiviral cell state) [27,28] in relation to the transcriptional factors Zink Finger and SCAN, which play a fundamental role in cell differentiation and proliferation [29]. There were also matches between the viral glycoprotein precursor (with a structural function in the viral adsorption and fusion steps) [30] and several isoforms (1–8) of the human protein Golgin B1 (which has functions of RNA binding and intermediation of compartments between the Golgi apparatus and the Endoplasmic Reticulum) [31]. SOUV also showed protein matches with the eight isoforms of Golgin B1, but the corresponding viral side was polymerase, which has the function of manipulating and replicating RNA [32]. Machado et al. (2018) ruled out the association of Golgin B1 gene polymorphisms and cleft palate, but considering the in silico evidence presented, the involvement of Golgin B1 in the pathogenesis of CL/P should not be ruled out [33]. Unlike its taxon relatives, LCMV did not show amino acid matches in the analysis, with most of its points being due to nucleotide matches. Used as a model species in the study of arenaviruses, LCMV is also associated with severe fetal damage when congenital infection occurs. Unlike the other members of Baltimore class V, arenaviruses have the capacity for ambivalent transcription in their ssRNA−, allowing for simultaneous polypeptide translation in opposite directions. LCMV replication occurs strictly in the cellular cytoplasm, and its nucleoprotein blocks the intracellular immunological activity mediated by IFN-I (Type I Interferon). Budding needs to be mediated by viral protein Z, which, in turn, interacts with several cellular proteins to direct the viral particle to the cytoplasmic membrane, restarting the cycle in another cell. Transmission occurs via fomites, contact with other infected animals, and vertical transmission with high fetal viremia. Arenavirus infections are common; however, severe cases with manifestations of hemorrhagic fevers and meningitis can occur [34].

Two of the seven G5 outliers belong to the *Orthomyxoviridae* family: *Betainfluenzavirus influenzae* (FLUVB) and *Gammainfluenzavirus influenzae* (FLUVC) [15]. With a TPV of 104, FLUVB showed amino acid matches between its polymerase and Microtubule-Associated Protein 7, which plays an essential role in cell differentiation [31], while FLUVC, with a TPV of 117, obtained matches between viral polymerase and Collagen Chain Alpha 1 (XII). It is relevant to mention that Alpha 1 Collagen Chain (XI) is associated with cleft palate [4] and that the difference between Alpha 1 Collagen Chain (XII) and Alpha 1 Collagen Chain (XI) is only in the extracellular matrix according to cell type, both being associated fibrils (binding short structures) [35]. Acs et al. (2005) associated cases of CL/P with influenza viruses [36] but without specifying the species; considering the evidence presented, we can infer that these were cases of FLUVB and/or FLUVC. Orthomyxoviruses have segmented genomes; in the case of FLUVB and FLUVC, there are eight and seven genomic segments, respectively, guaranteeing, in these viruses, the possibility of viral rearrangement through segment exchange when the host cell is infected by different viruses. This dynamism in terms of the evolution of orthomyxoviruses favors the increase in host range and the potential for their dissemination in reservoir hosts [37].

*Ledantevirus kern* (KCV), which showed a TPV of 113, and *Vesiculovirus perinet* (PERV), with the second highest TPV (232), are members of the *Rhabdoviridae* family [15]. One of the most interesting protein matches of PERV was between its matrix protein (with the main function of controlling viral RNA transcription, in addition to inactivating host transcription and mediating cytopathic processes such as apoptosis) [38] and Receptor Type A Ephrine (involved in cell migration and organization) [39]. Considering the previous finding, it is relevant to mention that Nasreddine, El Hajj, and Ghassibe-Sabbagh (2021) presented a possible association between Type B1 Ephrine Receptor and the pathogenesis of CL/P [4]. The rhabdoviruses PERV and KCV infect mammals, with the genus *Ledantevirus* being associated with bats and possible transmission to other mammals via arthropods and the genus *Vesiculovirus* being associated with possible similar transmission, presenting flu-like semiology and leading to encephalitis [38].

G4 had *n* = 129 viruses (Appendix A) with a mean TPV of 11.4 (Table 2) and had four outliers (Table 3). Consisting of ssRNA+, the genome of these viruses is already arranged in the reading direction, and the synthesis of polyproteins, which are subsequently cleaved, guarantees several advantages against intracellular defenses [25].

*Human coronavirus NL63* (HCoV-NL63) and *Human coronavirus HKU1* (HCoV-HKU1) showed TPVs of 29 and 77, respectively; they are both members of the *Coronaviridae* family but belong to different genera, with the first being an *Alphacoronavirus* and the second a *Betacoronavirus* [15]. Only HCoV-HKU1 showed amino acid matches aligning its non-structural protein 3 (with different binding properties and resistance to host defenses) [40,41] with Afadin isoform 6 (protein signaling and cell junction) [31]. Another fact related to HCoV-HKU1 is that it is in the group of outliers with FLUVC. This is because, even if they belong to different Baltimore classes, as shown by Snijder et al. (1991), the two may present a possible occurrence of non-homologous RNA recombination; the above-mentioned suggested, in their conclusion, the possible participation of cellular mechanisms in this event [42]. Both HCoV-NL63 and HCoV-HKU1 are associated with diseases of the lower and upper respiratory tract and can also be transmitted congenitally [43,44].

A member of the genus *Alphavirus* [15], the outlier *Rio Negro virus* (RNV), had a TPV = 30. Its transmission via biological vector is common in mammals, and it is also relevant to remember that the ability of transplacental passage of alphaviruses is known [45]; so far, RNV has not been associated with human diseases. 

The last outlier to be described for G4 is *Cosavirus B* (CoSV-B), which presented TPV = 38. A member of the *Picornaviridae* family and genus *Cosavirus* [15], this virus has been detected in animals and humans, sick and healthy, in fecal samples and pharyngeal swabs, with humans and pigs as natural hosts [46]. When detailing their amino acid alignments, its polyprotein showed matches with Forkhead Box F2; Nasreddine, El hajj, and Ghassibe-Sabbagh (2021) presented two proteins from the same family as being associated with cleft palate [4]. 

It is important to emphasize that the study by Silva, Messias, and Soares (2022) [12] which showed the etiological potential of flaviviruses (G4) to cause CL/P, is corroborated by the results of the present study, as G4 showed a statistically significant difference in relation to G1, G3, and G5. However, we must emphasize that the sampling scenario, in the current study, is general in terms of reference and known viruses that infect the *H. sapiens* species, contrary to the study mentioned above, which used a specific approach for flaviviruses. 

With an average TPV = 166.9, G3 (Appendix A) showed significance in relation to G4 and G2 (Table 3). Baltimore class III viruses have dsRNA genomic organization, which means that when they enter the cytoplasm, RdRp transcribes messenger ribonucleic acids for the synthesis of structural and non-structural proteins that form the viral progeny. The use of its ssRNA− part serves as the basis for transcription in the viral factory (cytoplasmic location), dictating the molecular course according to the demands of biosynthesis. If we compare this strategy to that of G5 viruses, which have only ssRNA− genomic organization [25], we can hypothesize that the presence of negative-sense RNA strands favors the pathogenesis of CL/P. G3 includes the participation of four species of rotaviruses (Appendix A). In their study, Vieira, Pereira, and Carvalho (2010) suggest the importance of gastroenteritis caused by these agents (without specifying the viral species) in neonates with craniofacial malformations, but their findings do not consider these viruses as an etiology of malformations [47]. There are no studies (or at least they have not been found) that investigate the transplacental passage of rotaviruses; this might be because even though it is a common etiological agent of gastroenteritis among children (with oral–fecal transmission), its various variants (thanks to the recombination of genomic segments) remain infectious for humans, even in adulthood, though asymptomatically [48], so investigating the possibility of congenital infection would not be such an obvious hypothesis to be discarded. Therefore, G3, despite demonstrating significance in Dunn’s test, did not present outliers, which may be related to the evident etiological potential for CL/P of all its members; we can infer that the TPV is not related to the statistical differences among Baltimore classes.

G2, with *n* = 72 (Appendix A) and TPV = 13 (Table 2), presented four outliers. Baltimore class II viruses are made up of ssDNA; the most distant characteristic compared with the other groups is perhaps the need to use the host cell’s DNA polymerase when it enters the S phase and thus replicate its genetic material and synthesize the necessary proteins for biosynthesis [25]. 

Two G2 outliers belong to the *Anelloviridae* family [15]: *Alphatorquevirus homin9* (TTV12), with TPV = 36, and *Torque teno mini virus 4* (TTMV4), with TPV = 41. Anelloviruses are associated with the possibility of playing roles in the balance of the host’s defense system in relation to diseases caused by other agents. Thanks to the great diversity of species and identified transmissive routes (air, blood, feces, and saliva), the co-infection of torque teno virus and torque teno mini virus has already been evidenced [49]. Among the two anelloviruses that were atypical, TTMV4 showed protein similarity with precursor protein isoforms of lysyl oxidase, which is involved in binding collagen and elastin in the extracellular matrix [50].

*Brisavirus* (HuRaBV) was one of the G2 outliers. A member of the *Redondoviridae* family, HuRaBV, fits into the etiological context of CL/P because the same was evidenced in oral samples. Its definitive host is not known, and there are no associated human diseases, although high loads of viral DNA have been found in cases of periodontitis [15,51]. 

*Bocaparvovirus primate2* (HBoV2c), a member of the subfamily *Parvovirinae* and genus *Bocavirus*, presented TPV = 73. Even though they are described as rare etiological agents in causing disease, bocaviruses are commonly transmitted via the fecal–oral cycle [15]. At the amino acid level, this virus showed similarities with the centromere cellular proteins, which is in line with the general strategy of the Baltimore class to which it belongs (in relation to the need to use DNA polymerase during the S cell phase) [25]. It is relevant to highlight the research study by Tiessen et al. (1994), who related CL/P with a fellow taxon (*Parvovirinae*) of HBoV2c, in this case, *Primate erythroparvovirus 1* (B19V) [52]; however, this virus did not show statistical significance in our analysis, with TPV = 13. Considering the absence of current studies relating B19V and CL/P, we can infer that Tiessen and his colleagues (1994) were limited by serological tests [52], considering the non-specificity of cross-reactions in the viral serological diagnosis [53], even more so in species related viruses such as B19V and HBoV2c [15].

The analyzed Baltimore class I viruses (Appendix A) also showed significance in relation to G2 and G4. These data may affect the theory of the ssRNA− x CL/P relationship in three possible ways: (A) These data nullify the hypothesis of the relationship of ssRNA− and CL/P. Alternatively, (B) the type of acid (deoxyribonucleic or ribonucleic) is not relevant in terms of the potential to cause CL/P, but the reading direction (3–5) and thus the biosynthesis strategy that involves genomic storage in complementary strands are. At the same time or alternatively, (C) one or more unknown factors (cellular and/or viral) in common in Baltimore classes I, III, and V and different in classes II and IV increase the potential to cause CL/P in relation to the other viruses.

G1, in addition to showing significance in Dunn’s test, had 5 outliers and 10 viruses that integrated EVEs into human sequences associated with CL/P. The viruses belonging to Baltimore class I have dsDNA, and most of its members replicate their genomes in the cell nucleus. However, poxviruses are an exception, as they do not use the cell nucleus [25], and deserve to be highlighted in the findings. Two outliers and five species associated with the integration of EVEs were evidenced, probably due to non-homologous recombination and/or interaction with cellular retroelements; we consider the latter case to be a greater possibility in recognition of the fact that its replication, as far as it is known, does not occur in the cell nucleus [10,54]. 

The poxviruses relevant to the context of this research study were *Monkeypox virus* (MPXV), *Volepox virus* (VPXV), *Abatino macacapox virus* (AMV), *Akhmeta virus* (AKMV), *Camelpox virus* (CMLV), *Taterapox virus* (TATV), all members of the genus *Orthopoxvirus*, and *Molluscum contagiosum virus* (MCV) of the genus *Molluscipoxvirus* [15].

Among the orthopoxviruses, MPXV (TPV = 86) and VPXV (TPV = 87) presented atypical samples but did not integrate any EVEs associated with CL/P into the host genome. However, the opposite occurred with AMV (TPV = 63), AKMV (TPV = 34), CMLV (TPV = 64), and TATV (TPV = 62), which fixed EVEs and did not present atypical TPVs. Nonetheless, all orthopoxviruses that fixed EVEs did so with reference to human sequences corresponding to Kelch-like family member 4. It is theorized that this group acts through its repeat regions as actin bonds; however, the function of this group has not yet been determined [31]. Considering the DNA hybridization ability of orthopoxviruses [15,54], we can hypothesize that at least one EVE fixation event occurred, which could have been disseminated via hybridization to other species. As for the question of which species initiated this fixation, our research group will try to answer these questions in future studies. Our estimated time of fixation of these EVEs disagrees with that reported in the study by Babkin, Babkina, and Tikunova (2022) [55], who inferred that the speciation of the first ancestor of the genus *Orthopoxvirus* occurred approximately 78,000 years ago. It is important to remember that the aforementioned study is specific to viral evolution (considering only conserved sequences of viral genomes). The estimation of the time of EVE fixation does not follow the same methodology; for this reason, our results include more spaced time intervals, as they are based on host speciation and not the number of substitutions (nucleotides or amino acids) x time. Recalling our estimates, the only orthopoxvirus with possible more recent EVE fixation was AKMV (less than 6 million years ago), in contrast to its taxon partners with a fixation time over 16 million years ago. With this, we could assume that the AKMV ancestor has a greater chance of not having integrated EVEs but receiving the corresponding sequence from its older relatives via hybridization. Therefore, our estimates are not specifically about these viral species but about their ancestors, viruses that are probably already extinct, but which passed their “fossil” records of their possible relationship with CL/P to their current progeny.

MCV, on the other hand, showed EVE fixation at a different time, over 16 million years ago, related to the BEN domain-containing protein, with a crucial function in cellular DNA processing and replication in eukaryotes [56]. Mulluscipoxviruses were among the first genera to speciate in the *Chordopoxvirinae* subfamily [15]. Considering the principles of coevolution, virus and host [10], and the findings presented, we can infer that the identified EVE fixation time portrays the beginning of the unique and efficient biosynthesis strategies of poxviruses and their relationship with the host species’ current data; however, a more robust study is needed to better support this hypothesis. An EVE integrated less than 6 million years ago is related to cAMP and cAMP-inhibited cGMP 3′,5′-cyclic phosphodiesterase 10A, associated with the function of controlling transcriptional signaling through cyclic control nucleotides [57].

*Gammapapillomavirus 7* (HPV109) showed an atypical TPV of 88. *Alphapapillomavirus 11* (HPV34) did not show an outlier; nevertheless, it was probably responsible for EVE fixation. Neither HPV109 nor HPV34 is related to any specific semiology, unlike the other viruses in the same family, which can vary as etiological agents of skin lesions, usually involving cellular transformation [58]. HPV109 showed a similarity between the amino acids of its E1 protein (which, when together with E2, mediates viral DNA replication and release) [59] and olfactory receptor 2B2. The integrated EVEs of HPV34 were in nucleotide sequences of HIVEP zinc finger 1 (with the cellular and viral function of transcriptional regulation) [31]. Two sequences were integrated into the human genome by HPV34, one over 16 million years ago and the other between 6 and 8 million years ago. The latter was recent if we consider the approximate time of the oldest ancestor of papillomaviruses (~424 million years ago) [60]. These integrations probably occurred via non-homologous recombination [10] since papillomaviruses have a biosynthesis mechanism that is closer to that of the classic Baltimore class I model than poxviruses [25].

With the third highest TPV (118), *Betapolyomavirus hominins* (BKPyV) showed protein matches between agnoprotein (related to virion entry and exit) [61] and protein eyes shut homolog, a protein of the extracellular matrix with several unknown aspects related to its function but associated in many ways with other matrix proteins [62]. BKPyV, in addition to the common transmission of polyomaviruses (skin–skin and oral/fecal/urine transmission), can also be transmitted through the air (isolated from respiratory aspirates) and water. The congenital route of transmission has already been hypothesized [63]. 

The evidence also points to the potential of the *Herpesviridae* family in the etiology of CL/P, considering that the largest TPV (704) in this research study was related to *Cytomegalovirus humanbeta5* (HHV-5), in addition to having fixed EVEs along with the related *Varicellovirus humanalpha3* (HHV-3) and *Roseolovirus humanbeta7* (HHV-7). The estimated emergence of the ancestor of herpesviruses that we know today occurred approximately 400 million years ago [64]. These viruses are very well adapted to their hosts, being able to capture genes and even develop similar ones through evolutionary convergence, which is the case of several immunomodulatory factors and enzymes in their genome that came from cellular genes [65]. This corroborates the evidenced EVEs; HHV-5 introduced a sequence in the region corresponding to Phosphodiesterase 7B (over 16 million years ago) responsible for mediating several transcription signals [31], which corroborates well the strategy of latency and active cycle of herpesviruses [66]. HHV-7, on the other hand, fixed EVEs at two different times (between 6 and 8 million years ago and less than 6 million years ago), referring to solute carrier family 35 member F1, which has an inferred function in transmembrane active transport [31]. As for HHV-3, it integrated EVEs over 16 million years ago, also referring to the nucleotide region that encodes the protein eyes shut homolog mentioned in the previous paragraph in relation to BKPyV. Considering the potential of herpesviruses as CL/P etiological factors is not unprecedented [67,68,69,70,71], with three representatives (*Simplexvirus humanalpha1* (HHV-1), HHV-2, and HHV-5) standing out among the microorganisms that make up the acronym STORCH (Syphilis, Toxoplasmosis, Rubella, Cytomegalovirus, and Herpes), i.e., agents considered in the literature to cause congenital malformations [72]. The data here presented only reaffirm a potential in herpesviruses that cannot be ruled out, i.e., their potential indirectly related to CL/P, which has possibly been built and outlined in millions of years of coevolution between the ancestors of the viruses used here as samples and the ancestors of *H. sapiens*.

As a limitation, in this study, we only used reference samples (sequences) of viral species, and this may not represent the true scenario (for example, the exclusion of G7 from the analysis due to sample insufficiency), when we consider that viruses that we do not know can infect our species and that even viral members of the same species can show significant differences. The study of variants and strains could increase the robustness of these data; however, the time taken to perform this would not be feasible, considering that the object of study is the most abundant and variable organism on the planet [10,73]. This limitation in the study of viruses also occurs in other methodologies, such as serology and the possibility of cross-reactions, molecular detection, viral load variation in the sample, and detectability of viral load in the sample type [53]. The etiological investigation of CL/P, because it is a disease with a complex and multifactorial etiological character, contrary to many diseases caused by a solely viral etiology, leads to discovering one piece of the puzzle at a time, which is further complicated by the absence of multidisciplinary research. 

Therefore, the results of the present study are relevant as they corroborate some results in other research methodologies as discussed previously, as they provide the basis for more specific studies with the same and/or other approaches that could reveal a so far scarcely the investigated relationship. This score-based system can also be adapted to other scenarios, different viral groups, and complex etiologies in which a possible relationship is hypothesized. 

## 5. Conclusions

In this in silico study, we deduced that viruses hold potential as etiological factors for CL/P. Irrespective of their biosynthesis class, it is crucial to pay significant attention to 20 viral species within the etiological context; consequently, public health protocols associated with CL/P should be revised accordingly. Furthermore, with reference to the Baltimore classification, we demonstrated an association between classes I, III, and V and classes II and IV. We also inferred an indirect coevolutionary history that links viruses with CL/P, indicating a connection with herpesviruses, papillomaviruses, and poxviruses that could date over sixteen million years. 

## Figures and Tables

**Figure 1 viruses-16-00511-f001:**
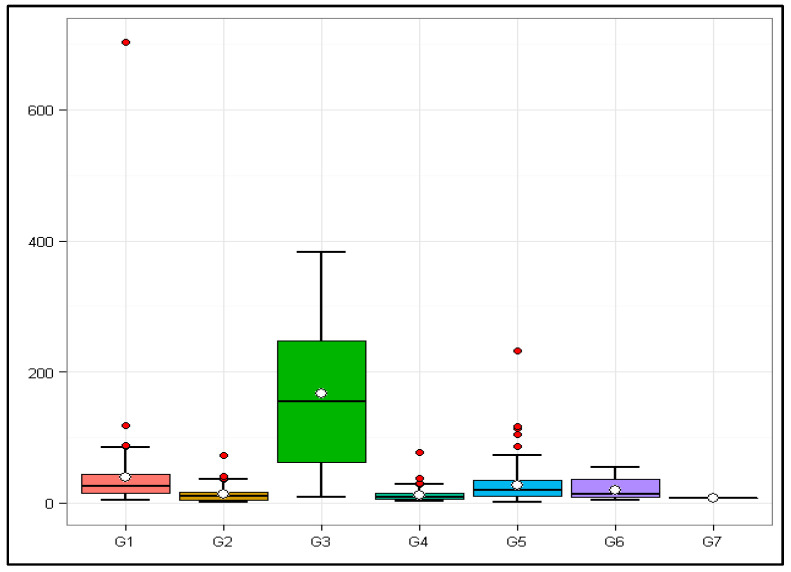
Boxplot of sample groups according to the Baltimore scale and the respective Total Potential Values of their members. Horizontal line—Sample groups; Vertical line—Total Potential Values of members of the sample groups; Red dots—Outliers; G1: Baltimore class I; G2: Baltimore class II; G3: Baltimore class III; G4: Baltimore class IV; G5: Baltimore class V; G6: Baltimore class VI; G7: Baltimore class VII.

**Figure 2 viruses-16-00511-f002:**
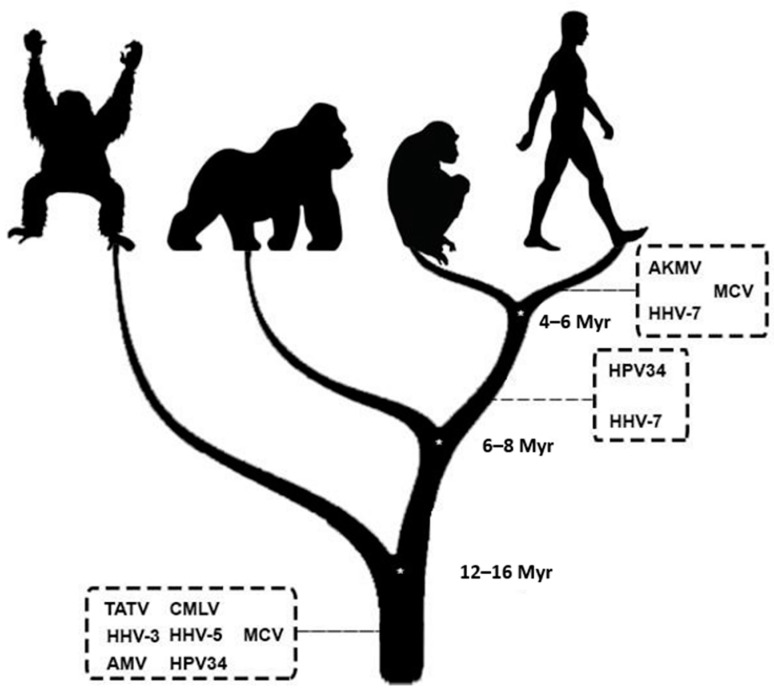
Evolutionary temporal position of the viruses that fixed Endogenous Viral Elements in human sequences related to Orofacial Clefts. Representative vectors from left to right—Orangutan, Gorilla, Chimpanzee, and Human; Myr—Millions of years; AKMV—*Akhmeta virus*; MCV—*Molluscum contagiosum virus*; HHV-7—*Roseolovirus humanbeta7*; HPV34—*Alphapapillomavirus 11*; TATV—*Taterapox virus*; CMLV—*Camelpox virus*; HHV-3—*Varicellovirus humanalpha3*; HHV-5—*Cytomegalovirus humanbeta5*; AMV—*Abatino macacapox virus*; The time of the dendrogram nodes is based on the work of Pääbo (2003) [19].

**Table 1 viruses-16-00511-t001:** Score-based system adapted from Silva, Messias, and Soares (2022) [12] with the addition of amino acid similarity investigation and Endogenous Viral Elements.

	Topic	Question	Response	Score
**I**	**Transplacental transmission**	Can the virus cross the placental barrier?	If negative	0
			If positive	2
			If never investigated/not found	1
**II**	**Virus and orofacial cleft** **associations**	Are there any studies in the literature in which the virus is associated with orofacial clefts?	No/not found	0
			Positive (semiology)	1
			Positive (serology)	2
			Positive (polymerase chain reaction and/or viral isolation)	3
**III**	**Nucleotide similarity**	Does the virus have nucleotide similarity with human sequences associated with orofacial clefts? (BLAST 2.13.0)	For each match with E-value ≤ 1	+1
**IV**	**Amino acid similarity**	Does the virus have amino acid similarity with human sequences associated with orofacial clefts? (BLAST 2.13.0)	For each match with E-value ≤ 0.05 obtained in topic III and subject to protein translation	+1
**V**	**Presence of Endogenous Viral Elements (EVEs)**	Could the virus possibly result in EVE integration into a human sequence associated with orofacial clefts? (BLAST 2.13.0)	For each match with an E-value ≤ 0.0001 obtained in topic III resubmitted in alignment with a database with general nucleotide sequences and with a positive result for the same virus of origin (method adapted from Katzourakis and Gifford (2010)) [11]	+1
**TPV**	**Total Potential Value (TPV)**	What is the numerical value of the potential of the virus to cause orofacial clefts?	Sum of values obtained in previous topics	X

**Table 2 viruses-16-00511-t002:** Descriptive statistics of the sample groups according to the Baltimore scale.

Group	N	Median	1º Quartile	3º Quartile
G1	91	25	15	43
G2	72	10,5	4	16.75
G3	7	154	62	248
G4	129	8	6	15
G5	137	19	10	35
G6	5	13	8	35.5

N: Sample number; G1: Baltimore class I; G2: Baltimore class II; G3: Baltimore class III; G4: Baltimore class IV; G5: Baltimore class V; G6: Baltimore class VI.

**Table 3 viruses-16-00511-t003:** Dunn’s statistical method of the sample groups according to the Baltimore scale.

Comparison	Difference in Ratings	Test Q	*p*-Value
G3 vs. G4	239 vs. 638	4.845	<0.001
G3 vs. G2	234 vs. 366	4.645	<0.001
G3 vs. G6	177 vs. 329	2.376	0.262
G3 vs. G5	133 vs. 502	2.703	0.103
G3 vs. G1	87 vs. 632	1.753	1
G1 vs. G4	152 vs. 006	8.712	<0.001
G1 vs. G2	146 vs. 734	7.299	<0.001
G1 vs. G6	89 vs. 697	1.532	1
G1 vs. G5	45 vs. 87	2.661	0.117
G5 vs. G4	106 vs. 136	6.788	<0.001
G5 vs. G2	100 vs. 865	5.437	<0.001
G5 vs. G6	43 vs. 827	0.755	1
G6 vs. G4	62 vs. 309	1.073	1
G6 vs. G2	57 vs. 037	0.968	1
G2 vs. G4	5 vs. 272	0.281	1

G1: Baltimore class I; G2: Baltimore class II; G3: Baltimore class III; G4: Baltimore class IV; G5: Baltimore class V; G6: Baltimore class VI; Difference in ratings: Measuring the Total Potential Value of which virus, the comparative relationship between the groups, and generating a numerical value as a result; Test Q: Critical value to consider whether the difference is significant or not.

**Table 4 viruses-16-00511-t004:** Significant outliers in relation to their Baltimore classes.

Viral Species and Abbreviation	Baltimore Class	Total Potential Value
*Human coronavirus NL63* (HCoV-NL63)	IV	29
*Rio Negro virus* (RNV)	IV	30
*Alphatorquevirus homin9* (TTV12)	II	36
*Brisavirus* (HuRaBV)	II	38
*Cosavirus B* (CoSV-B)	IV	38
*Torque teno mini virus 4* (TTMV4)	II	41
*Bocaparvovirus primate2* (HBoV2c)	II	73
*Human coronavirus HKU1* (HCoV-HKU1)	IV	77
*Monkeypox virus* (MPXV)	I	86
*Mammarenavirus machupoense* (MACV)	V	86
*Volepox virus* (VPXV)	I	87
*Souris mammarenavirus* (SOUV)	V	87
*Gammapapillomavirus 7* (HPV109)	I	88
*Betainfluenzavirus influenzae* (FLUVB)	V	104
*Lymphocytic choriomeningitis mammarenavirus* (LCMV)	V	105
*Ledantevirus kern* (KCV)	V	113
*Gammainfluenzavirus influenzae* (FLUVC)	V	117
*Betapolyomavirus hominis* (BKPyV)	I	118
*Vesiculovirus perinet* (PERV)	V	232
*Cytomegalovirus humanbeta5* (HHV-5)	I	704

**Table 5 viruses-16-00511-t005:** Endogenous Viral Elements fixed by a Baltimore class I viruses in human sequences associated with Orofacial Clefts.

Viral Species and Abbreviation	Endogenous Viral Element	Maximum Fixation Time
*Varicellovirus humanalpha3* (HHV-3)	ACTCTCTCTCTTTCTCtatatatatatatatatata	Over 16 million years
	AACTCTCTCTCTTTCTCtatatatat-atatatatatat	
	tatatatatatatatatataGAGAAAGAGAGAGAGT	
	atatatatatat-atatatataGAGAAAGAGAGAGAGTT	
*Cytomegalovirus humanbeta5* (HHV-5)	TCGTCCTCTTCCTCTTCTTCCTCCTCTTC	Over 16 million years
*Roseolovirus humanbeta7* (HHV-7)	CATATATTTGCACATACTAATGTGTTCATGTGGGTATATGTACATAT-TACATA---ATATATGCTAGTAAATGATTACATGCACTAGCATATATTTGCACATACTAATGTGTTCATGTGGGTATATGTACATAT-TACATA	6 to 8 million years old
	TGGGTATATGTACATATTACATAATATATGCTAGTAAATGATTACATGCACTAGCATATATTTGCACATACTAATGT—GTTCATGTGGGTATATGTACATATTACATAATATATGCTAGTAAATGATTACATGCACTAGCATATATTTGCACATACTAATGTGTTCATGTGGGTATATGTACATAT	
	TGGGTATATGTACATATTACATAATATATGCTAGTAAATGATTACATGCACTAGCATATATTTGCACATACTAATGT—GTTCATGTGGGTATATGTACATAT	Under 6 million years
	CATATATTTGCACATACTAATGTGTTCATGTGGGTATATGTACATAT-TACATA	
	ATATATTTGCACATACTAATGTGTTCATGTGGGTATATGTACATAT	
*Alphapapillomavirus 11* (HPV34)	TGTTAAAAGTATATATATTATATGTGTGTGTGTTT-TATAT	Over 16 million years
	GTATATATAT---TATATGTGTGTGTGT-TTTATAT	6 to 8 million years old
	GTATATATATTATATGTGTGTGTGT-TTTATAT	
*Molluscum contagiosum virus* (MCV)	GCACCTGCGCAAGATGTACG-GCGCAAGCAGGTCTATCTACAACTTCGCTGTGCGTATGC TCGTGTACATGTTTCCAGAGCTCTTTACTGCGGAGAACCTGCACACGCACTTCAACTGCT ACGGCTCCATGGGCAAGC-GCAGGCTCGACCCGCTACGCCTGCGCTTGCTCCGGCACTAT GTGCAGTTGCTGCACCCGGCGGCGCGC----AACGAGCGCGTGTGGATCACAAAGTTCCT GGCGTGCCTGGACGAGCGCTGCCGGCGCCGCTGCGCACGGA-CACAGGCGC	Over 16 million years
	TCCTCTCCCGGGGAGCTGGCGGTGCTCCTACTGCACAAGGTCTTCCAGGAGCTCTTTGAC GCGCGCCAGCTGCGCCGCTGCTACAGCTGCTACGGCGACGGGCGCACGCATTGTCTGGAC CCTGCGCGCCTGCAACTGATCCGGCACTGCGTGGCGCTTTGCTTCCCTTCCATG	
	GATGAGGGAA----Atgtgtgtgtgtgtgtgtgtgtgtgtgtgtg	Under 6 million years
	cacacacacacacacacacacacacacacaT----TTCCCTCATC	
*Abatino macacapox virus* (AMV)	GGACGCATTTATGTTATCGGTGGACGAGATGGATCAAATTATCTAAACACTGTAGAAAGTTGGAAACCT-ATGGACAACAAGTGGCAATACG	Over 16 million years
*Akhmeta virus* (AKMV)	GGACGCATTTATGTTATTGGTGGACGAGATGGATCAAATTATCTAAACACTGTAGAAAGTTGGAAACCT	Under 6 million years
	GGACGCATTTATGTTATTGGTGGACGAGATGGATCAAATTATCTAAACACTGTAGAAAGTTGGAAACCT	
*Camelpox virus* (CMLV)	ATTTATGTTATCGGTGGTCGAGATGGATCAAATTATCTAAACACTGTAGAAAGTTGGAAACCT	Over 16 million years
*Taterapox virus* (TATV)	ATTTATGTTATCGGTGGTCGAGATGGATCAAATTATCTAAACACTGTAGAAAGTTGGAAACCT	

## Data Availability

The original contributions presented in the study are included in the article/Appendix A, further inquiries can be directed to the corresponding author/s.

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
