# Peer review of "Potential of Viruses as Environmental Etiological Factors for Non-Syndromic Orofacial Clefts"

_viruses, 2024, doi:10.3390/v16040511_

Round 1

Reviewer 1 Report

Comments and Suggestions for Authors

The aim of the present article was to analyze the potential for viral infections in the species Homo sapiens to serve as an environmental cause of orofacial clefts (OFC).

From where were the viruses gathered?

Is the figure original?

Could table 5 be strinked in order to fit on one page because like this it is verry hard to follow?

In the discussion i would suggest the authors to include more recent published articles.

The counclusion should be just one paragraph.

It is too big.

Comments on the Quality of English Language

Moderate

Reviewer 2 Report

Comments and Suggestions for Authors

The aim of this study was to analyze the potential for viral infections in the species Homo sapiens to serve as an environmental cause of orofacial clefts. The topic of the study is interesting.

The study has a good methodology and is well written. The statistical analyses used in the study are adequate.

The results are clearly presented.

In the Discussion the Authors discussed the obtained results and compared with other findings. The limitations of the study were also discussed. This part of the study could be written more concisely.

Please prepare the reference list according to the Instructions for Authors.

Reviewer 3 Report

Comments and Suggestions for Authors

This is an in silico consideration of the possibility of viral involvement in the aetiology of orofacial clefts.

There is consideration of five domains (topics) of information.

The first two are assessed by searching four databases (lines 89-91) but no detail is given about the searches (e.g PECO statement as anchor for the search), inclusion/exclusion criteria, data extraction, assessment of risk of bias and process for integrating evidence (not meta-analysis but there appears to have been a qualitative assessment of links with the other three domains, and a quantitative one based on scores for each domain)

The other three domains were assessed using bioinformatics databases, with two of them being conditional on the domain of nucleotide similarity. Is this dependence the reason for the use of the Dunn method? That does not seem well explained

in the consideration of human sequences associated with orofacial clefts, was there any consideration of tissue type for domains/topics III-V?

in topic II, case reports and comparative studies should be distinguished. 

lines 113-4: reference is made to “subject to availability”. Please explain how availability might have affected the results

Table 3: what is rationale for order? How should we interpret correlation coefficients ranging from 5 to 239 - these are very different from other types of correlation coefficients which typically range from -1 to +1   How should Q be interpreted

What about subtypes of OFC, eg syndromic vs. Non-syndromic, CP vs CL+-P?

it would have been helpful to document whether this approach detects associations that have been reported in observational studies in humans (epidemiological and genetic association), e.g PVRL1 gene variants and possible link with herpes (Romitti P et al, 2001), IRF6, Zika virus (guimaraes GM et al. Int j Oral Max Surgery 2019; 48:S1), influenza (Dymanus IL et al  FACE 2021; 2(1): 23-9.

Comments on the Quality of English Language

Lack of detail on methods as mentioned above

mosquito-borne viruses is keyword but it isn’t apparent why in reading text

discussion seems much too long

lines 358-361 , 377-80 unclear

line 496  what is cabalistic evidence?